Mitochondrial DNA in human identification: a review

Amorim António 1 2
Fernandes Teresa 3 4
Taveira Nuno ntaveira@ff.ul.pt 5 6
1 Instituto Nacional de Medicina Legal e Ciências Forenses , Lisboa , Portugal
2 Faculdade de Ciências da Universidade de Lisboa , Lisboa , Portugal
3 Escola de Ciências e Tecnologias, Universidade de Évora , Évora , Portugal
4 Research Center for Anthropology and Health (CIAS), Universidade de Coimbra , Coimbra , Portugal
5 Instituto Universitário Egas Moniz (IUEM) , Almada , Portugal
6 Research Institute for Medicines (iMed.ULisboa), Faculty of Pharmacy, Universidade de Lisboa , Lisbon , Portugal
Calafell Francesc
Electronic publication date: 2019 Aug 13
Publication date: 2019
Volume: 7
Electronic Location ID: e7314
Received 2019 Jan 23; Accepted 2019 Jun 18
Copyright: ©2019 Amorim et al.
Copyright year: 2019
Copyright holder: Amorim et al.
License: This is an open access article distributed under the terms of the Creative Commons Attribution License, which permits unrestricted use, distribution, reproduction and adaptation in any medium and for any purpose provided that it is properly attributed. For attribution, the original author(s), title, publication source (PeerJ) and either DOI or URL of the article must be cited.
License URL: https://creativecommons.org/licenses/by/4.0/

Keywords: mtDNA, Human identification, Legal medicine, Forensic biology

Funding: The authors received no funding for this work.

==============================
Mitochondrial DNA (mtDNA) presents several characteristics useful for forensic studies, especially related to the lack of recombination, to a high copy number, and to matrilineal inheritance. mtDNA typing based on sequences of the control region or full genomic sequences analysis is used to analyze a variety of forensic samples such as old bones, teeth and hair, as well as other biological samples where the DNA content is low. Evaluation and reporting of the results requires careful consideration of biological issues as well as other issues such as nomenclature and reference population databases. In this work we review mitochondrial DNA profiling methods used for human identification and present their use in the main cases of humanidentification focusing on the most relevant issues for forensics.

Introduction

Human genetic identification for forensic purposes is achieved through the definition of genetic profiles. A genetic profile or the genetic fingerprint of an individual is the phenotypic description of a set of genomic loci that are specific to that individual. In accordance with international recommendations, particularly with recommendations of the European DNA Profiling Group (EDNAP), currently, only genetic profiles obtained from autosomal short tandem repeats (STR) should be used for genetic fingerprinting. However, in a considerable number of situations of human identification, autosomal DNA is highly degraded or isn’t available at all. In these cases the study of mitochondrial DNA (mtDNA) for human identification can be the last appeal and by that reason has become routine (Budowle et al., 2003). Nevertheless, and despite the robustness of mtDNA in cases of exclusion or absence of identity between sequences, when the results are sequence identity, contrary to nuclear markers, they do not refer to an individual but to a group of individuals of the same maternal lineage.

Survey methodology

We systematically searched with PubMed Advanced Search Builder for papers titles with the following combinations: (1) mitochondrial DNA and biology, (2) mitochondrial DNA and guidelines, (3) mitochondrial DNA and nomenclature, (4) mitochondrial DNA and sequencing, (5) mitochondrial DNA and database, (6) mitochondrial DNA and data, (7) mitochondrial DNA and identification, (8) mitochondrial DNA and forensic. Papers nonrelated with human or animal mitochondrial DNA were excluded. Our search was not refined by publishing date, journal or impact factor of the journal, authors or authors affiliations. In addition, we used Guideline documents from the International Society for Forensic Genetics available at https://www.isfg.org/.

Mitochondrial DNA biology and genetics

Mitochondria are cellular organelles that contain an extrachromosomal genome, which is both different and separate from the nuclear genome. The mitochondrial DNA (mtDNA) was first identified and isolated by Margit Nass and Sylvan Nass in 1963, who studied some mitochondrial fibers that according to their fixation, stabilization and staining behavior, appeared to be DNA related (Nass & Nass, 1963). However, the complete sequence of the first mtDNA was only published and established as the mtDNA Cambridge Reference Sequence (CRS) eighteen years later, in 1981 (Anderson et al., 1981).

Essentially, the mtDNA is a five mm histone-free circular double-stranded DNA molecule, with around 16 569 base-pairs and weighting 107 Daltons (Taanman, 1999). mtDNA strands have different densities due to different G+T base composition. The heavy (H) strand encodes more information, with genes for two rRNAs (12S and 16S), twelve polypeptides and fourteen tRNAs, while the light (L) strand encodes eight tRNAs and one polypeptide. All the 13 protein products are part of the enzyme complexes that constitute the oxidative phosphorylation system. Other characteristic features of the mtDNA are the intronless genes and the limited, or even absent, intergenic sequences, except in one regulatory region.

The mitochondrial D-loop is a triple-stranded region found in the major non-coding region (NCR) of many mitochondrial genomes, and is formed by stable incorporation of a third 680 bases DNA strand known as 7S DNA (Kefi-Ben Atig et al., 2009). The origin of replication is located at the non-coding or D-loop region, a 1 121 base pairs segment that is located between positions 16 024 and 576, according to the CRS numeration (Anderson et al., 1981) (Fig. 1). The D-loop region, also comprehends two transcription promotors, one for each strand. Nucleotide positions in the mtDNA genome are numbered according to the convention presented by Anderson et al. (1981), which was slightly modified by Andrews et al. (1999), determining the replacement of CRS for rCRS (revised Cambridge Reference Sequence). More precisely, the numerical designation of each base pair is initiated at an arbitrary position on the H strand, which continues thereafter and around the molecule for approximately 16 569 base pairs.

Figure 1 The human mitochondrial DNA genome with genes and control regions labeled.

Adapted from Picard, Wallace & Burelle (2016).

The apparent lack of mtDNA repair mechanisms and the low fidelity of the mtDNA polymerase lead to a significant higher mutation rate in the mitochondrial genome, when compared to the nuclear genome. For example, Sigurğardóttir and collaborators, estimated the mutation rate in the human mtDNA control region to be 0.32 ×10−6/site/year (Sigurðardóttir et al., 2000) which compares to 0.5 × 10 −9/site/year in the nuclear genome (Scally, 2016). Most of the sequence variation between individuals is found in two specific segments of the control region, namely in the hypervariable region 1 (HV1, positions 16 024 to 16 365) and in the hypervariable region 2 (HV2, positions 73 to 340) (Greenberg, Newbold & Sugino, 1983). A third hypervariable region (HV3, positions 438 to 574), with additional polymorphic positions can be useful in the resolution of indistinguishable HV1/HV2 samples (Lutz et al., 2000). The small size and relatively high inter-person variability of the HV regions are very useful features for forensic testing purposes.

The mtDNA sequence defines the individual haplotype which is reported by the different base pairs relative to the rCRS mtDNA sequence. The collection of similar haplotypes defined by the combination of single nucleotide polymorphisms (SNPs) in mtDNA inherited from a common ancestor defines an haplogroup which was formed as a result of the sequential accumulation of mutations through maternal lineage (Mitchell et al., 2015).

A mitochondrion contains 2 to 10 copies of mtDNA and each somatic cell can have up to 1,000 mitochondria (Elson et al., 2001; Wei et al., 2017). Hence, when the amount of the extracted DNA is quite small or degraded, it is more likely that a DNA typing result can be obtained by typing the mtDNA than by typing polymorphic regions that are found in nuclear DNA.

Contrarily to the nuclear DNA, the mtDNA is exclusively maternally inherited, which justifies the fact that, apart from mutation, mtDNA sequence of siblings and all maternal relatives is identical (Case & Wallace, 1981; Giles et al., 1980; Hutchison et al., 1974). This specific characteristic can be very helpful in forensic cases, such as in the analysis of the remains of a missing person, where the known maternal relatives can provide some reference samples for a direct comparison to the mtDNA type. Due to the lack of recombination, maternal relatives from several generations apart from the source of evidence (or biological material) can be used for reference samples (Case & Wallace, 1981; Giles et al., 1980; Hutchison et al., 1974).

The haploid and monoclonal nature of the mtDNA in most individuals simplifies the process of interpretation of the DNA sequencing results. Still, it is possible to find heteroplasmy in occasional cases (Bendall, Macaulay & Sykes, 1997; Bendall & Sykes, 1995; Comas, Paabo & Bertranpetit, 1995; Gill et al., 1994; Ivanov et al., 1996; Wilson et al., 1997). A person is considered as heteroplasmic if she/he carries more than one detectable mtDNA type. There are two classes of heteroplasmy, related to length polymorphisms and to point substitutions. Only the latter is important for forensic human identification. Most forensic laboratories worldwide do not report length polymorphisms and the guidelines on human identification with mtDNA do not point them as mandatory information (Parson et al., 2014; Prinz et al., 2007). Furthermore, the information of length polymorphisms has no impact in haplogroups’ definition.

Heteroplasmy manifests itself in diverse ways (Stewart et al., 2001). An individual may show more than one mtDNA type in a single tissue. An individual may be heteroplasmic in one tissue sample and homoplasmic in another one. Finally, an individual may exhibit one mtDNA type in one tissue and a different type in another tissue. Of the three possible scenarios, the last one is the least likely to occur. When heteroplasmy is found in the mtDNA of an individual, it usually differs at a single base, in HV1 or HV2.

Heteroplasmy was observed at position 16,169 of the mtDNA control region in the putative remains of Tsar Nicholas II of Russia and his brother, the Grand Duke of Russia Georgij Romanov (Gill et al., 1994; Ivanov et al., 1996). Comas, Paabo & Bertranpetit (1995), in turn, detected heteroplasmy at two distinct positions, 16, 293 and 16,311, in the mtDNA of an anonymous donor’s plucked hair. Wilson et al. (1997) found a family constituted by a mother and two children carrying a heteroplasmic mtDNA at position 16,355 both in blood and buccal swab samples.

The existence of heteroplasmic individuals and the limited knowledge about both the mechanism and the rate of heteroplasmy can be issues raised in an attempt to exclude mtDNA evidence from forensic investigations. Heteroplasmy at one nucleotide position is more frequently observed in hair samples, mainly due to genetic drift and to bottlenecks which occur due to the hair follicle’s semiclonal nature (Budowle et al., 2003; Buffoli et al., 2013; Paus, 1998; Rogers, 2004). Hence, if an evidentiary hair sample contains one of the two heteroplasmic lineages that are observed in a reference sample, or vice versa, then the interpretation of exclusion may be incorrect. In this case, typing additional hairs may be required to solve the problem (Budowle et al., 2003).

As it was previously pointed out, it is accepted that the mitochondrial genome is maternally inherited. Even though the sperm contains a few mitochondria in the neck and in the tail region, the male mitochondrial genome is destroyed either during or shortly after the fertilization. More precisely, sperm mitochondria disappear in the early embryogenesis, either by selective destruction, inactivation or dilution (Cummins, Wakayama & Yanagimachi, 1997; Shitara et al., 2000; Shitara et al., 1998; Wallace, 2007). Nonethless, in the last years some cases of biparental mtDNA inheritance have been reported. In the most recent case, Luo and collaborators describe biparental mtDNA inheritance, either directly or indirectly, in 17 members of three multigenerational families, with results confirmed by two independent laboratories (Luo et al., 2018). Besides this report in humans, there are also a few examples of paternal inheritance of the mitochondrial genome in animals (Gyllensten et al., 1991), and by that reason and despite the limited evidence for paternal inheritance of the mitochondrial genome in humans, the courtroom can be tempted to use such possibility to depreciate the use mtDNA evidences.

Mitochondrial DNA Nomenclature

Considering that listing more than 600 bases in order to describe the results from a new HV1 and HV2 sequence would be unpractical, an alternative approach was developed which essentially identifies and reports the differences relative to the reference sequence rCRS (Anderson et al., 1981).

Even though the process of naming mtDNA sequences seems simple and obvious, it is crucial to properly consider the nomenclatures, since complications might arise. Most ambiguities in the alignment/nomenclature arise due to insertions and/or deletions (indels). To avoid ambiguities, facilitate haplotype identification and their assignment to existent haplogroups or to new haplogroups, phylogenetic-based nomenclature guidelines have been proposed. The phylogenetic approach provides an evolutionary based view of global mtDNA diversity that is scientifically sound because all mtDNA lineages derive from a common maternal ancestor. The phylogenetic notation of mtDNA haplogroups can be based on maximum parsimony or maximum likelihood analysis of mutations present in sequences from the control region or in mitogenomes (for comprehensive reviews in phylogenetic reconstruction methods see (Bianchi & Liò, 2007; De Bruyn, Martin & Lefeuvre, 2014). The most comprehensive repository of mtDNA genomes is Phylotree (http://www.phylotree.org) a website that also provides the reference phylogenetic tree describing the worldwide human mitochondrial DNA variation (Van Oven & Kayser, 2008). The phylogenetic tree shown in Phylotree is regularly updated with new haplogroups as found by maximum parsimony analysis of new mtDNA haplotypes using the mtPhyl software (https://sites.google.com/site/mtphyl/home) (Van Oven, 2015). A maximum likelihood approach for mtDNA haplogroup classification named EMMA has been recently described by Röck and collaborators (Röck et al., 2013).

Variants flanking long C tracts are subject to sequence-specific conventions. The long C tracts of HVS-I and HVS-II should always be scored with 16 189C and 310C, respectively. Length variation of the short A tract preceding 16 184 should be notated preferring transversions unless the phylogeny suggests otherwise. Regarding deletions, these are recorded by the number of the base(s) that is missing, with respect to the rCRS, followed by DEL or del or (-) (for example 249 DEL or 249 del or 249-). When any of the four bases are observed, N notation should be used. Indels should be placed 3′ with respect to the light strand unless the phylogeny suggests otherwise. For example, if the bases beyond the position 309 were out of the register by one base due to the insertion of a C the mutation is designated as 309.1C. Two C insertions are designated as 309.1C and 309.2C. Important tools to assist with the notation of mtDNA sequences are available at http://empop.org/. These notations are used for storing haplotypes in the EMPOP database and have also been adopted by the Scientific Working Group on DNA Methods (SWGDAM) in the United States (Parson et al., 2014).

Overall, the large majority of individuals from African populations, and specially from sub-Saharan African populations, are categorized into one of the main haplogroup lineages that diverged from macro-haplogroup L: L0, L1, L2, L3, L4, L5 and L6 (Allard et al., 2005; Bandelt et al., 2001; Behar et al., 2008; Chen et al., 2000; Gonder et al., 2007; Pakendorf & Stoneking, 2005; Rosa et al., 2004; Van Oven & Kayser, 2008). On the other hand, more than 90% of the individuals of the European and USA Caucasian populations are categorized into 10 main haplogroup lineages: H, I, J, K, M, T, U, V, W and X (Allard et al., 2002; Behar et al., 2008; Budowle et al., 2003; Pakendorf & Stoneking, 2005; Torroni et al., 1996). Concerning to African-American populations, the most commonly observed haplogroups are L2a, L1c, L1b and L3b (Allard et al., 2005). The main haplogroups found in individuals from Asian populations are haplogroups M and N (Allard et al., 2004; Kivisild, 2015).

Mitochondrial DNA Typing Guidelines

In 2014 the DNA Commission of the International Society of Forensic Genetics (ISFG) published updated guidelines and recommendations concerning mitochondrial DNA typing. These guidelines referred to good laboratory practices, targeted region, amplification and sequencing ranges, reference sequence, alignment and notation, heteroplasmy, haplogrouping of mtDNA sequences, and databases and database searches. In Table 1 we present the 16 recommendations of ISFG. Overall, these are the main guidelines concerning the application of mtDNA polymorphisms in human identification, which are regularly revised and published by the International Society of Forensic Genetics (Bär et al., 2000; Parson et al., 2014; Prinz et al., 2007; Tully et al., 2001).

Table 1 Guidelines of the DNA Commission of the International Society of Forensic Genetics, 2014.

Addressement	Recommendation	Statement	
General recommendations/ good laboratory practice	Recommendation #1	Good laboratory practice and specific protocols for work with mtDNA must be followed in accordance with previous guidelines	
	Recommendation #2	Negative and positive controls as well as extraction reagent blanks must be carried through the entire laboratory process	
	Recommendation #3	Reported consensus sequences must be based on redundant sequence information, using forward and reverse sequencing reactions whenever practical	
	Recommendation #4	Manual transcription of data should be avoided and independent confirmation of consensus haplotypes by two scientists must be performed	
	Recommendation #5	Laboratories using mtDNA typing in forensic casework shall participate regularly in suitable proficiency testing programs	
Targeted region, amplification and sequencing ranges	Recommendation #6	In population genetic studies for forensic databasing purposes, the entire mitochondrial DNA control region should be sequenced.	
Reference sequence	Recommendation #7	MtDNA sequences should be aligned and reported relative to the revised Cambridge Reference Sequence (rCRS, NC001807), and should include the interpretation range (excluding primer sequence information)	
Alignment and notation	Recommendation #8	IUPAC conventions using capital letters shall be used to describe differences to the rCRS and (point heteroplasmic) mixtures. Lower case letters should be used to indicate mixtures between deleted and non-deleted (inserted and non-inserted) bases. N-designations should only be used when all four bases are observed at a single position (or if no base call can be made at a given position). For the representation of deletions, “DEL”, “del” or “S” shall be used	
	Recommendation #9	The alignment and notation of mtDNA sequences should be performed in agreement with the mitochondrial phylogeny (established patterns of mutations). Tools to assist with the notation of mtDNA sequences are available at http://empop.org/	
Heteroplasmy	Recommendation #10	In forensic casework, laboratories must establish their own interpretation and reporting guidelines for observed length and point heteroplasmy. The evaluation of heteroplasmy depends on the limitations of the technology and the quality of the sequencing reactions as well as the experience of the laboratory. Differences in both PHP and LHP do not constitute evidence for excluding two otherwise identical haplotypes as deriving from the same source or same maternal lineage	
	Recommendation #11	For population database samples, length heteroplasmy in homopolymeric sequence stretches should be interpreted by calling the dominant variant, which can be determined by identifying the position with the highest representation of a non-repetitive peak downstream of the affected stretch	
Haplogrouping of mtDNA sequences	Recommendation #12	MtDNA population data should be subjected to analytical software tools that facilitate phylogenetic checks for data quality control. A comprehensive suite of QC tools is provided by EMPOP	
Databases and database searches	Recommendation #13	The entire database of available sequences should be searched with respect to the sequencing (interpretation) range to avoid biased query results	
	Recommendation #14	Laboratories must be able to justify the choice of database(s) and statistical approach used in reporting	
	Recommendation #15	Laboratories must establish statistical guidelines for use in reporting an mtDNA match between two samples	
	Recommendation #16	Highly variable positions such as length variants in homopolymeric stretches should be disregarded from searches for determining frequency estimates. Heteroplasmic calls should be queried in a manner that does not exclude any of the heteroplasmic variants	

Mitochondrial DNA Sequencing Methodologies

In 1977 Sanger presented the first DNA sequencing technology (Sanger, Nicklen & Coulson, 1977), also called the chain termination method and now known as first generation sequencing. The incorporation of ddNTPs in newly synthesized DNA strands results in termination of the elongation process and correspondent knowledge about the specific nucleotide present at the sequence at each position. Sanger sequencing method can produce reads from 25 up to 1200 nucleotides, allowing the read of a maximum of 96 kb nucleotides in 2 h.

Since 2005 new sequencing methods, also known as next generation sequencing (NGS) methods, have been developed (Bruijns, Tiggelaar & Gardeniers, 2018). Sequencing by synthesis methods such as Roches’ 454 Pyrosequencing, and Illuminas’ HiSeq, allow sequencing up to 80 million base pairs in 2 h or up to 6 billion base pairs in 1–2 weeks (Mascher et al., 2013; Pukk et al., 2015). Sequencing by hybridization and ligation, such as the ABI SOLiD 3plus platform, yields 60 gigabases of usable DNA data per run. With these massive parallel sequencing (MPS) technologies, sequenced DNA fragments can range from 35-75 nucleotides as in the SOLiD technology (Bruijns, Tiggelaar & Gardeniers, 2018; Ondov et al., 2010; Shendure et al., 2005), to 100–1,000 nucleotides as in 454 pyrosequencing (Bruijns, Tiggelaar & Gardeniers, 2018; Dames et al., 2010). Ion Torrent’s Personal Genome Machine™ (PGM) uses a detection methodology based in pH change upon addition of a nucleotide to a sequence (Rothberg et al., 2011). When this happens protons are released generating an electric signal that is proportional to the amount of protons released. Data collection is carried out by a complementary metal-oxide semiconductor (CMOS) sensor array chip with the sensor surface present at the bottom of the well plate, and these chips can measure millions to billions of simultaneous sequencing reactions (Liu et al., 2014). Finally, MinION (Oxford Nanopore Technologies), a portable real-time sequencing device, allows ultra-long read lengths (hundreds of kb) albeit with lower accuracy (Oikonomopoulos et al., 2016).

The NGS technologies have been quickly applied in forensics (Bruijns, Tiggelaar & Gardeniers, 2018). For example, Ion Torrent’s PGM system has been used for sequencing complete mitogenomes (Parson et al., 2013) and to study heteroplasmy (Magalhães et al., 2015) in the forensic context. Although PGM proved to be sensitive and accurate at detecting and quantifying mixture and heteroplasmy, there were some problems in the coverage of the mtDNA genome with some regions presenting extreme strand bias, and presenting false positives mostly generated by alignment problems in the analysis algorithms. More recently Ion S5 System (Thermo Fisher Scientific) and MiSeq FGx Desktop Sequencer (Illumina) were used to evaluate the Precision ID mtDNA Whole Genome Panel (Woerner et al., 2018). Both sequencing systems provided consistent estimation of mtDNA haplotypes. Many other studies on the use of NGS technologies for forensic genetics and mtDNA analysis have been published (Chaitanya et al., 2015; Churchill et al., 2018; Hollard et al., 2017; Just, Irwin & Parson, 2015; Just et al., 2014a; Just et al., 2014b; Lopopolo et al., 2016; Ma et al., 2018; Marshall et al., 2017; Ovchinnikov et al., 2016; Park et al., 2017; Templeton et al., 2013; Young et al., 2017). However, further validation studies and specialized software functionality tailored to forensic practice should be produced in order to facilitate the incorporation of NGS processing into standard casework applications (Amorim & Pinto, 2018; Peck et al., 2016). In the meantime, and according to current international guidelines (Parson et al., 2014; Prinz et al., 2007), Sanger sequencing still continues to be an adequate method for mtDNA analysis for forensic human identification, and is used in most casework laboratories worldwide (Ballard, 2016). Some forensic laboratories perform Sanger sequencing for HVI and HVII fragments, while others have already extended the study to the HVIII fragment and, in recent years, most of the forensic laboratories are introducing the amplification of the entire control region as routine methodology (Chaitanya et al., 2016; Poletto et al., 2019; Turchi et al., 2016; Yasmin et al., 2017). Attempting to improve the power of mtDNA in human identification, over the past decade some studies have been focused in the extension of the analyses to the whole mtDNA genome (Duan et al., 2019; Strobl et al., 2018; Woerner et al., 2018). Nevertheless, it should be stressed that while the information from the entire mtDNA genome can contribute to refine the haplogroup obtained with the study of HVI, HVII and HVII fragments or the entire control region, it is not supposed to change the previous results to a different haplogroup of a completely different geographic ancestry.

Mitochondrial DNA Population Data and Databases

When two mtDNA sequences, one from an evidence sample and another from a reference sample, cannot be excluded as being originated from the exact same source, it is necessary to convey some information concerning the rarity of the mtDNA profile. The current practice is to count how many times a specific sequence is observed within a population database(s) (Budowle et al., 1999). Overall, the population databases that are used in forensics comprehend several convenience samples, representing the major population groups of the potential contributors in terms of evidence.

The most important mtDNA haplotypes database is the EDNAP Mitochondrial DNA Population Database (EMPOP, http://www.empop.org) (Parson & Dür, 2007). In its early stages, EMPOP was designed and envisioned to serve as a reference population database, specifically to be used in the evaluation of the mtDNA evidence around the world, aiming to provide the highest quality mtDNA data. The architecture of this online database and its analysis tools have evolved over the last few years, even though the main emphasis of the EMPOP database remains to be mtDNA data quality. Therefore, and as a direct consequence, EMPOP not only serves as a reference population database, but also as a quality-control tool for scientists in forensic genetics, as well as in other disciplines. Finally, and even though there is a significant number of high-quality reference population databases for forensic comparisons, EMPOP is the most comprehensive resource, especially from the standpoint of the populations that are represented in such database (Parson et al., 2014).

EMPOP uses SAM, a string-based search algorithm that converts query and database sequences into alignment-free nucleotide strings and thus guarantees that a haplotype is found in a database query regardless of its alignment. SAM-E, an updated version of SAM that considers block InDels as phylogenetic events, is used currently. At EMPOP, the tool haplogroup browser represents all the established Phylotree haplogroups in convenient searchable format and provides the number of EMPOP sequences assigned to the respective haplogroups by estimating mitochondrial DNA haplogroups using the maximum likelihood approach EMMA (Röck et al., 2013). For multiple possible haplogroups, most recent common ancestor (MRCA) haplogroups are provided.

As mentioned before, PhyloTree provides an updated comprehensive phylogeny of global human mtDNA variation, based on both coding and control region mutations (Van Oven & Kayser, 2008). The complete mtDNA phylogenetic tree includes previously published as well as newly identified haplogroups, is continuously and regularly updated, and is available online at http://www.phylotree.org. At EMPOP the geographical haplogroup patterns are provided via maps to visualize and better understand their geographical distribution (Fig. 2).

Figure 2 Representation of the geographical origin of the main mtDNA haplogroups, based on Lott et al. (2014).

Another important human mtDNA database is Mitomap (Ruiz-Pesini et al., 2007). In 1996, this database developed into an online database (http://www.mitomap.org) containing published human mtDNA variation along with geographic and disease specific variants. Currently, Mitomap is manually curated, frequently updated and a functionally rich resource, presenting high-quality human mtDNA data for clinicians, investigators and geneticists (Ruiz-Pesini et al., 2007). Mitomap has three main categories for usage. It contains some background information regarding the human mitochondrial DNA, such as the general representation of mtDNA, haplogroups and their frequencies and illustrations of mtDNA, among others. Furthermore, users can also find information about other mtDNA-specific databases, tools and useful resources.

Mitomap stores the annotated listing of the mtDNA variants from both healthy individuals and patients. The frequencies of the variants are calculated from human mitogenomes retrieved from the GenBank. Therefore, users can retrieve information about the loci, the nucleotide change, the codon position and the number, among others, and download the most important data in different file formats.

Mitomap contains the Mitomaster analysis tool, currently providing the Application Programming Interface for it. The main function of this tool is to allow the identification of polymorphic positions, the calculation of variant statistics and the assignment of haplogroups to complete or partial mitogenomes. Such query might be performed by recurring to mtDNA sequences, to GenBank identifiers or to single nucleotide variants (Brandon et al., 2009).

From another perspective, ethical and legal problems may arise in the implementation of mtDNA databases. The informative potential which the analysis of mtDNA entails can generate privacy questions (Guillen et al., 2000; Wallace et al., 2014). Mitochondrial diseases affect between 1 in 4,000 and 1 in 5,000 people. In most people, primary mitochondrial disease is a genetic condition that can be inherited. Information about the mitochondrial genome composition may therefore enable the identification of the current or future state of health of an individual. For this reason, the analysis of mtDNA must be carried out only on non-coding regions, which have not been associated with any kind of disease or phenotypical information.

Mitochondrial DNA in Forensic Human Identification

In the context of forensic analysis, both mtDNA sequences of a reference sample and an evidence sample(s) are compared. When the sequences are unequivocally different, the conclusion is that they can be excluded as being originated from the same source. Although not stated in any research paper or guideline text, forensic routine laboratories tend to accept as an exclusion scenario when more nucleotide differences exist between the two sequences. If the mtDNA sequences are identical, the samples cannot be excluded since they must have the same origin or derive from the same maternal lineage. Similarly, samples can’t be excluded when heteroplasmy is observed at the same nucleotide positions in both samples. When one sample is heteroplasmic and the other is homoplasmic but they both share at least one mtDNA species, the samples can’t be excluded since they may have the same origin. Several authors have suggested that samples with mtDNA with one-base difference should be further evaluated, mainly regarding their rate of mutation (Alonso et al., 2002; Bär et al., 2000; Holland & Parsons, 1999; Parson et al., 2014; Tully et al., 2001).

In this section, we present some selected published cases of human identification with mtDNA. Table 2 summarizes the selected published cases. Stoneking et al. (1991) presented the first report of successful application of the mtDNA typing to a case that involved the individual identification of skeletal remains. This was the case of a 3-year-old child disappeared from her parents’ house in October of 1984. In March of 1986, the skeletal remains of a human child were found in the desert, 2 miles away from the parents’ residence. Using hybridization with 23 sequence-specific oligonucleotide probes (SSO) targeting nine regions of HV1 and HV2 on the control region, they found that the skeletal sample and the mother shared the same mtDNA types, corroborating that those skeletal remains were of the missing child. Moreover, they anticipated that the mtDNA typing would be valuable not only in linking biological remains to missing individuals, but also in the analysis of material in sexual assault cases.

In July of 1990, the body of a female, in a quite advanced state of decomposition, was discovered in an open field. Despite being impossible to identify the remains by analyzing the individual’s clothes and fingerprints, her dentition was consistent with old dental records of a missing person from the same region. Some fragments of the heel bone and fibula, plus samples of the hair and skin, were provided for the DNA analysis, as well as a blood sample from a putative sister of the deceased. Sullivan, Hopgood & Gill (1992) attempted the identification of the highly decomposed remains of the corpse, amplifying and directly sequencing 2 hypervariable segments within HV1 and HV2 in the mtDNA. No statistical value was given to the evidence, since no database of the British population sequences were available at that time. Still, no differences were found between both sequences, the blood of the putative sister and the bone of the corpse, indicating they were sisters.

Table 2 Selected published cases of human identification with mtDNA.

Reference/Year	Studied samples	mtDNA studied regions	Used methodologies	Reference samples	Results	
Stoneking M, Hedgecock D, Higuchi RG, Vigilant L, Erlich HA. Population variation of human mtDNA control region sequences detected by enzymatic amplification and sequence-specific oligonucleotide probes. Am J Hum Genet. 1991;48(2):370–82.	Skeletal remains of a human child, found in 1986	HVI, HVII	PCR for amplification
Hybridization with oligonucleotide probes for sequence determination	Parents of a 3-year-old child disappeared from home in 1984	Identical mtDNA sequence in skeletal remains and sample of the 3-year-old child mother
Positive ID	
Sullivan KM, Hopgood R, Gill P. Identification of human remains by amplification and automated sequencing of mitochondrial DNA. Int J Legal Med. 1992;105(2):83–6.	Body of a female, in an advanced state of decomposition discovered in 1990	HVI, HVII	PCR for amplification
Sanger sequencing	Blood sample from a sister of a deceased female at the same region	No differences were observed between the corpse and blood from the putative sister
Positive ID	
Gill P, Ivanov PL, Kimpton C, Piercy R, Benson N, Tully G, et al. Identification of the remains of the Romanov family by DNA analysis. Nat Genet. 1994;6(2):130–5.	Nine skeletons found in a grave in Ekaterinburg, Russia, 1991	HVI, HVII	PCR for amplification
Sanger sequencing	Blood sample from Gt. Gt. Grandson of Louise of Hesse-Cassel and from Gt. Gt. Gt. Granddaughter of Louise of Hesse-Cassel	Exact sequence between putative Tsarina Alexandra and putative three children.
Exact mtDNA results between putative Tsar Nicholas II and two living maternal relatives of the Tsar	
Ivanov PL, Wadhams MJ, Roby RK, Holland MM WV& PT. Mitochondrial DNA sequence heteroplasmy in the Grand Duke of Russia Georgij Romanov establishes the authenticity of the remains of Tsar Nicholas II. Nat Genet. 1996;(12):417–20.	Skeleton of putative Tsar Nicholas II	HVI, HVII	PCR for amplification
Sanger sequencing	Skeleton of Grand Duke of Russia Georgij Romanov (Tsar’s brother)
Blood sample from Countess Xenia Cheremeteff-Sfiri (maternal Tsar’s relative)	Establishment of the authenticity of the remains of Tsar Nicholas II	
Deng YJ, Li YZ, Yu XG, Li L, Wu DY, Zhou J, et al. Preliminary DNA identification for the tsunami victims in Thailand. Genomics, Proteomics Bioinforma. 2005;3(3):143–57.	258 tooth samples from killed people at the 2004 Southeast Asia Thailand Tsunami	HVI, HVII	PCR for amplification
Sanger sequencing	200 relatives of the tsunami victims	200 tsunami victims have been identified, including both Thai nationals and foreign tourists from several nations	
Ríos L, García-Rubio A, Martínez B, Alonso A, Puente J. Identification process in mass graves from the Spanish Civil War II. Forensic Sci Int. 2010;219(1–3).	Skeletal remains exhumed from a mass grave from the Spanish Civil War (1936–1939)	HVI, HVII	PCR for amplification
Sanger sequencing	Sister of the youngest person presumptively known to be buried in the grave	Match between mtDNA profiles of the biologically youngest skeleton and the sister of the youngest person presumptively known to be buried in the grave	
Piccinini A, Coco S, Parson W, Cattaneo C, Gaudio D, Barbazza R, et al. World war one Italian and Austrian soldier identification project: DNA results of the first case. Forensic Sci Int Genet. 2010;4(5):329–33.	Remains of missing soldiers occasionally found during excavations	HVI, HVII	PCR for amplification
Sanger sequencing	Offspring of the italian soldier Libero Zugni Tauro	Both mtDNA and Y-STR data showed clear exclusion scenarios
between the human remains and the reference samples	
King TE, Fortes GG, Balaresque P, Thomas MG, Balding D, Delser PM, et al. Identification of the remains of King Richard III. Nat Commun. 2014;5:1–8.	Skeleton excavated at the presumed site of the Grey Friars friary in Leicester, 2012	Whole mitochondrial genome	PCR for amplification
Massive parallel sequencing	Saliva samples of the modern relatives of Richard III	Positive mtDNA match between the only known female-line of Richard III and studied modern relatives of Richard III	
Ossowski A, Diepenbroek M, Kupiec T, Bykowska-Witowska M, Zielińska G, Dembińska T, et al. Genetic Identification of Communist Crimes’ Victims (1944–1956) Based on the Analysis of One of Many Mass Graves Discovered on the Powazki Military Cemetery in Warsaw, Poland. J Forensic Sci. 2016;61(6):1450–5.	Remains of eight people buried in one of many mass graves, which were found at the cemetery Powazzki Military in Warsaw, Poland	HVI, HVII	PCR for amplification
Sanger sequencing	Reference material was collected from the closest living relatives of Communist Crimes’ Victims (1944–1956)	Positive mtDNA match between 6 putative victims and 6 living relatives	
Ambers AD, Churchill JD, King JL, Stoljarova M, Gill-King H, Assidi M, et al. More comprehensive forensic genetic marker analyses for accurate human remains identification using massively parallel DNA sequencing. BMC Genomics. 2016;17(Suppl 9).	Human skeletal remains with 140-year-old discovered at a historical site in Deadwood, South Dakota, United States	HVI, HVII and ten fragments of mtDNA coding region	PCR for amplification
Massive parallel sequencing	Not used	Results were consistent with previous anthropological report that points to a male of European ancestry	

Perhaps the most well-known lineage study using mtDNA sequencing is related to the identification of Tsar Nicholas II’s bones. Gill et al. (1994) and Ivanov et al. (1996) compared the sequences of HV1 and HV2 fragments of the mtDNA obtained from the putative bones of the Tsar with those of Tsar living maternal relatives, Countess Xenia Cheremeteff-Sfiri and the Duke of Fife. It was found that the sequences were very similar, corroborating the hypothesis that the bone remains were of Tsar Nicholas II.

In a distinct scenario, Deng et al. (2005) used direct sequencing of the HV1 and HV2 fragments of the mtDNA control region to identify Tsunami victims in Thailand in 2004. This tsunami killed nearly 5,400 people in Southern Thailand, including foreign tourists and local residents. They succeeded in obtaining fully informative results for mtDNA markers (HV1 and HV2) from 258 tooth samples with a success rate of 51% (258/507).

More recently, Ríos, Ovejero & Prieto (2010) used direct sequencing of the HV1 and HV2 fragments of the mtDNA control region to identify human skeletal remains that were exhumed from a mass grave from the Spanish Civil War (1936–1939). There was a match between the mtDNA profiles of the biologically youngest skeleton and the sister of the youngest person that was presumptively known to be buried in the grave, allowing the identification of that person.

Also in 2010, Piccinini et al. (2010) attempted to identify the remains of a famous World War One Italian soldier that was killed in a battle along the Italian front in 1915. Like previous studies, they used the direct sequencing of the HV1 and HV2 fragments of the mtDNA control region to define single mtDNA haplotypes. The availability of the offspring maternal lineage allowed the mtDNA analysis, which presented a clear exclusion scenario: the remains did not belong to the supposed war hero.

In 2012, a skeleton was excavated at the site of the Grey Friars friary, in Leicester, which is the last-known resting place of King Richard III (King et al., 2014). To determine if the remains belonged to King Richard III, the HV1, HV2 and HV3 regions of the mtDNA of the skeletal remains and of the living relatives of King Richard III were sequenced and compared. There was a perfect match between the sequences indicating that the remains belong to King Richard III.

The communist period in Poland during 1944–1956 resulted in the death of more than 50,000 people, who were buried in secret. One mass grave was found at the cemetery Powazki Military, in Warsaw, Poland. In 2016, Ossowski and collaborators (Ossowski et al., 2016) identified 50 victims, specifically by using autosomal, Y-STR and direct sequencing of the HV1 and HV2 fragments of the mtDNA control region.

In 2016, among the first studies on human identification with mtDNA using massive parallel sequencing, Ambers et al. (2016) proposed a protocol that includes the study of ten regions of mtDNA for the identification of historical human remains with forensic genetic markers. They studied a 140-year-old human skeletal remains discovered at a historical site in Deadwood, South Dakota, United States. The remains were in an unmarked grave and there were no records available regarding the identity of the individual. The mtDNA profiles of the unidentified skeletal remains obtained with their method were consistent with H1 haplogroup. This haplogroup is the most common in Western Europe. The ancestry-informative nuclear SNPs also studied in this case indicated a European background. These genetic results are consistent with the findings of previous anthropological report which determined that the Deadwood unidentified skeletal remains belong to a male of European ancestry.

In 2017, the victims’ remains from the World Trade Center terrorism act, which occurred in September 11 of 2001, were still being identified by using the mtDNA sequencing technology, among other techniques, with protocols and guidelines as recommended by the International Society for Forensic Genetics (Goodwin, 2017).

Conclusions

Over the last 25 years, mtDNA typing has been widely used around the world to solve several human identification related issues in violent crimes, lesser crimes, acts of terrorism, mass disasters and missing persons’ cases. The progress in mtDNA typing has been overwhelming, going from the examination of small fragments in a matter of days to sequencing multiple entire mtDNA genomes in a couple of hours. Being a lineage genetic marker, mtDNA genome can provide information about ancestors, including health/disease information. Even though many would readily accept that there are good reasons for researchers to obtain information about an unknown suspect’s potential ancestral background, many still find the potential to determine genetic dispositions to certain disorders as being unacceptable. Hence, new technologies that enable mitogenome sequencing must be wisely used and for the reasons that they are intended, considering their specific focus and contribution within the field of forensic human identification.

Some concerns still remain regarding admissibility of mtDNA analysis in court especially related with the issue of heteroplasmy and, more recently, with the possibility of biparental inheritage. The complete elucidation of molecular mechanisms driving biparental inheritage of mtDNA, the ability to determine the situations where this is likely to occur, and the ability to identify and characterize heteroplasmy with high accuracy, are important issues that need to be addressed in order to ensure the robustness of mtDNA as an important and alternative tool in forensic human identification.

Additional Information and Declarations

Competing Interests

Author Contributions

Data Availability

The authors declare there are no competing interests.

António Amorim, Teresa Fernandes and Nuno Taveira conceived and designed the experiments, performed the experiments, analyzed the data, contributed reagents/materials/analysis tools, prepared figures and/or tables, authored or reviewed drafts of the paper, approved the final draft.

The following information was supplied regarding data availability:

This is a review article and did not collect or generate raw data.

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
