# Peer review of "Mitochondrial DNA in human identification: a review"

_PeerJ, doi:10.7717/peerj.7314_

## Round 0.1 · original submission · Major Revisions

Please address all the issues raised by the reviewers

Reviewer 1 ·

Basic reporting

The paper is an overview of mtDNA typing in forensics and the main topics of the field are illustrated and discussed, though not all of them in exhaustively way. The main issue is to better and clearly explains the fundamental concept concerning the use of mtDNA in forensic identification, that is mtDNA data cannot be applied to the identification of individuals in the same way that nuclear marker can, but it can be used only to make inference about the maternal lineage. This important point should be emphasized.
Other important areas to be improved and updated are nomenclature and sequencing methodology sections.

Experimental design

no comment

Validity of the findings

no comment

Additional comments

There are some points that should be revised:
Introduction
Lines 33, 39: it’s not very clear the difference between “medico-legal purposes” and “forensic purposes”. Please explain why you used both terms.
Line 41: references 1-4 are related to guidelines for mitochondrial DNA typing and they not seem very pertinent in this point of introduction. I suggest to mention other general reviews regarding the use of mtDNA in forensics.
Lines 42-50: This paragraph is identical to the abstract in the previous page. I suggest that you improve the introduction to better explain that mtDNA data cannot be applied to the identification of individuals in the same way that nuclear marker can, but it can be used to make inference about the maternal lineage. I suppose that this review is addressed to not expert users in forensic mtDNA, so this important point should be emphasized.
Survey methodology
Lines, 57-59: If not those cited, what are the selection criteria used?
Mitochondrial DNA biology and genetics
Line 84: Reference number 10 isn’t Andrews et al. Please correct. Moreover the revised mtDNA sequence (rCRS) published by Andrews et al. in 1999 is the reference sequence currently used. Therefore Authors should replace CRS with rCRS throughout the text (e.g lines 161 and 165).
Line 116: The term “haplogroups” need explanation and definition.
Lines 132-141: The sequences comparison issue seems to be not inherent in this paragraph. I suggest to put this issue in another specific paragraph related to forensic interpretation and reporting results of mtDNA typing. Moreover it could be appropriate to mention also the SWGDAM mtDNA interpretation guidelines 2013.
Mitochondrial DNA Nomenclature
Authors describe in this chapter the issue of mtDNA polymorphisms nomenclature, however the topic has not been exhaustively handled, it has poorly precise and it has not updated. For these reasons, this section requires a thorough and wide review. First of all, authors mentioned the nomenclature approach of Wilson et al (ref 37) based of a set of formal rules that was later modified by Budowle et al in 2010, but only the first version of this approach is describe in this review, so it isn’t update. Moreover, the ISFG proposed another kind of nomenclature, based on phylogenetic approach (see guidelines in ref. 4) and it has not even been mentioned. It would be appropriate to mention both and discuss the difference. Otherwise the Authors explain why describe only one approach. Actually, Authors mentioned the phylogenetic approach in Table 2, in the context of ISFG guidelines, but without a discussion. Moreover, the revised mtDNA sequence (rCRS) is the reference sequence used for comparison and notation of polymorphisms and not the CRS. Deletions are annotated as “DEL” or “-“ , and not with a “D”, to avoid confusing with IUB code, where D is considered a mixture of A, G, and T.
Mitochondrial DNA typing guidelines
Line 197: Table 1 is cited after Table 2…it could be better to rename tables in order of appearance in the text.
Mitochondrial DNA sequencing methodology
In this chapter Authors provide an overview of the DNA sequencing methods used in forensic, but I suggest to limit the text only to sequencing methods used for mtDNA typing. The text will appear more fluid and a more detailed description of Sanger sequencing by capillary electrophoresis, which is the most widely used method, would be useful. Moreover, the use of terms like “second generation”, “third generation” seems to be a little confused, please revised to provide more clear definitions. Lastly, the review lacks of information about mtDNA amplification strategies ( e.g complete mtGenome amplification, or entire control region (CR) amplification in a single PCR reaction, or CR region amplification in different overlapping fragments, or HVI-HVII-HVII in separate amplification) which could improve this chapter.
Lines 236-240: reference 53 could be cited together with other ref at line 246, as it isn’t one of the first study of mtDNA sequencing by MPS and it is relative to heteroplasmy issue.

Mitochondrial DNA population data and databases
The text needs a different organization. I suggest to move the EMPOP section after line 257, as it represent the most important mtDNA haplotypes database for forensic purposes, among those mentioned in the text.
Lines 258-269: I suggest to eliminate this part, as it seems to be not pertinent in this chapter, and to move it to another chapter.
Mitochondrial DNA population data and databases
Line 385: reference 97 isn’t relative to Park et al. and it isn’t cited in table 1. Please insert in table 1 reference numbers of cited papers.

Reviewer 2 ·

Basic reporting

The article is a review on the role of mitochondrial DNA in medico-legal area. In this regard, the article requires a thorough revision and the following issues should be adressed:
- No reference is made to the last article on the presence of paternal mitochondrial DNA in a group of families, which would have consequences in the field of forensic identification.
- It is mentioned CRS but not rCRS or RSRS., why?
- Reference is made to the nomenclature but the way of transcribing the mutations is not that used in the forensic field (line 162 16,192 C).
- There is some confusion with the reference 93. It is not clear if it refers to an article from 2012. However, in this case mitochondrial DNA is not used. It could be also an article of 2010 but there would be wrong authors, publication data, etc.
- Finally, the conclusions are somewhat surprising. The topic of recombination or paternal mitchondrial DNA is referred to as "challenges", however, this has not been addressed in the review. It also indicates that for years its admissibility or its relation to diseases has been questioned but this aspect are not adressed in the revision either.
In my opinion these are severe flaws of a wide scope review like this.

Experimental design

No coment

Validity of the findings

No coment

---

## Round 0.2 · Minor Revisions

Please correct the minor issues detected by reviewer 2

Reviewer 1 ·

Basic reporting

no comment

Experimental design

no comment

Validity of the findings

no comment

Additional comments

The authors appear to have addressed all the issues raised in review. The only point to solve is that Authors in their rebuttal letter stated that reference to Ambers et al. was included in Table 2, but is not so.

Reviewer 2 ·

Basic reporting

No comment

Experimental design

No comment

Validity of the findings

No comment

Additional comments

The authors have reviewed the text with all the corrections that were made to them. But the should check:
line 187. Regarding the nomenclature there are some errors. In the case of deletions, "del" is also accepted in addition to DEL or -. In addition N implies that the 4 bases appear and not that it is unambiguous. In addition, those rules that are literally copied from Parson's article should appear in quotation marks or in italics. The authors should review that paragraph.
line 348. The exclusion rules indicated do not correspond to the article by Parson et al., 2014. Indicate the correct reference.
References. Please change the Rios reference. The corresponding reference to Rios et al. (2010) is:
Identification process in mass graves from the Spanish Civil War I.
Ríos L, Ovejero JI, Prieto JP.
Forensic Sci Int. 2010 Jun 15;199(1-3):e27-36. doi: 10.1016/j.forsciint.2010.02.023.

---

## Round 0.3 · accepted · Accept

There seem to be no further issues to fix